# Lichen-Associated Oribatid Mites in the Taiga Zone of Northeast European Russia: Taxonomical Composition and Geographical Distribution of Species

Elena N. Melekhina 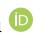

Institute of Biology, Komi Science Centre, Ural Branch of the Russian Academy of Sciences, (IB FRC Komi SC UB RAS), Kommunisticheskaya 28, 167982 Syktyvkar, Russia; melekhina@ib.komisc.ru

**Abstract:** We examined 35 species of ground and epiphytic lichens, including fruticose, foliose, and crustose lichen, as habitats of oribatid mites. Observations were carried out in the taiga forests of northeast European Russia, and 87 oribatid species from 38 families were found. The Crotoniidae, Carabodidae, Oppiidae, and Suctobelbidae are the most numerous families in ground lichens and the Oribatulidae are in the epiphytes. The families Micreremidae and Licneremaeidae were only noted in epiphytes. A complex of species characteristic of epiphytic lichens as habitats of oribatid mites have been identified, these are *Carabodes labyrinthicus*, *Oribatula* (Z.) *propinqua*, *Phauloppia nemoralis*, *Micreremus brevipes*, *Licneremaeus licnophorus*, *Furcoppia* (*Mexicoppia*) *dentata*, *Cymbaeremaeus cymba*. Only in epiphytes, rarely, the species were also *Jacotella frondeus*, *Ameronothrus oblongus*, *Mycobates* (*Calyptozetes*) *tridactylus*, and *Liebstadia humerata*. Characteristic for ground lichens are species *Trhypochthonius cladonicolus*, *Carabodes marginatus*, *Carabodes subarcticus*. Often found in both ground and epiphytic lichens are species *Eueremaeus oblongus* s. str., *E. oblongus silvestris*, *Ceratoppia quadridentata*, *Adoristes ovatus poppei*, *Graptoppia* (*Apograptoppia*) *foveolata*, *Suctobelbella acidens duplex*, *Tectocepheus velatus*, *Trichoribates berlesei*, *Chamobates pusillus*, *Diapterobates oblongus*, *Oribatula tibialis*, *Oribatula* (Z.) *exilis*, *Scheloribates laevigatus*, *Neoribates aurantiacus*, *Pergalumna nervosa*. In epiphytic lichens, we observed species that are rare in the North (*Oribatula* (Z.) *frisiae*, *O.* (Z.) *propinqua*, *P. nemoralis*, *L. licnophorus*, *F.* (*M.*) *dentata*, et al.), with some noted in the taiga zone for the first time (*J. frondeus*).

**Keywords:** oribatida; taxonomic diversity; distribution; checklist; fruticose lichen; foliose lichen; crustose lichen; ground lichens; epiphytic lichens





## 1. Introduction

Oribatid mites are a component of lichen consortia along with other invertebrate taxonomic groups [1–7]. Researchers have repeatedly noted that oribatids are permanent and numerous inhabitants of moss–lichen heaths [8–12], lichen and moss cover on trees [1,2,13–20], rocks [21,22], and coastal cliffs [23–26]. Lichens are often mentioned as the habitat of microarthropods [27–30]. Oribatid mites, inhabitants of epiphytic lichens, may represent bioindicators of radioactive contamination in natural ecosystems [31]. Complexes of oribatid mite species associated with lichens of different life forms have been identified as growing on different substrates in various plant communities [15–18,32–38].

For the taiga forests of northeast European Russia, we obtained data on the taxonomic composition, population structure, and distribution features of oribatid mites in lichens of different species and life forms [39,40]. The population structure of oribatid mites in lichens of six species, namely the ground *Cladonia arbuscula*, *C. rangiferina*, *C. stellaris*, *Cetraria islandica*, and the epiphytic *Hypogymnia physodes* and *Bryoria subcana* in four forest communities were discussed. It has been established that the species dominant in abundance in the ground lichens of the genus *Cladonia* were *Carabodes subarcticus*, *C. marginatus*, *Trhypochthonius cladonicolus*, *Scheloribates laevigatus*, and *Tectocepheus velatus*, while in *Cetraria islandica* the species were

*Adoristes ovatus poppei, C. subarcticus* and *T. velatus.* These were additionally associated with *Cetraria islandica* species *Ceratoppia quadridentata* and *Camisia biurus.* In epiphytes, the composition of dominant species was different, they were *Carabodes labyrinthicus, Oribatula (Z.) propinqua, Phauloppia nemoralis,* and *Diapterobates humeralis.* In total, 55 species of oribatid mites were found for the examined lichen species.

Species that have shown biotopic association with a series of habitats or a specific habitat have been identified. Based on the biotopic preferences of species and their abundance characteristics, we identified the ecological groups of oribatid species associated with epiphytic and ground lichens as habitats. Relative to the series of habitats investigated, five ecological groups of oribatid mites were named [41]. Two groups—arboricolous-dominant and arboricolous non-numerous—reflect the specificity of the species composition of oribatids in epiphytic lichens. The groups of hemiedaphic-dominant and hemiedaphic non-numerous species are characteristic of ground lichens. One group of arboricolous hemiadaphic species comprises species inhabiting both ground and epiphytic lichens. Ecological vicariate species related to epiphytes have been identified [41].

To date, we have conducted a study of the taxonomic composition of oribatid mites in lichens of 35 species of different life forms in the European North-East of Russia. The results have not yet been summarized. No analysis of the zoogeographic structure of the fauna was performed. This study aimed to summarize all the data obtained by the author on the taxonomic diversity of oribatid mites associated with lichens and to analyze information on the geographical distribution of species.

## 2. Materials and Methods

### 2.1. Research Area

This study was conducted in the taiga zone of northeast European Russia, in the Middle Taiga subzone, in the vicinity of the village of Kazhym, Koygorodsky District, Komi Republic (60°19′58″ N, 51°32′00″ E). The climate in the study area is moderately continental, with long cold winters and short cool summers [42,43]. The cold period of the year lasts 170–180 days. The average monthly temperatures in January and July are −15 and 17 °C, respectively. On some days, the temperature may drop to −45 °C when Arctic air invades. In summer, during short periods of tropical air, the temperature rises to 36 °C. Relative air humidity in winter months is 83–86%, and in the warm period of the year, it decreases to 53–60% [42]. The duration of the frost-free period is approximately 100 days, the annual amount of precipitation is up to 700 mm, and the average maximum height of snow cover in the forest is 100 cm [42,43]. In winter, the depth of frozen soil in the south of the republic is approximately 60 cm and more than 100 cm in the north.

The predominant plant formations are coniferous taiga phytocenoses. In the Middle Taiga subzone, spruce forests occupy the largest areas in the uplands, the tree layer of which consists of Siberian spruce *Picea obovata* Lebed. and sometimes contains a mixture of birch *Betula pubescens* and fir *Abies sibirica*. The most typical associations are blueberry and green-moss spruce forests; in the herb-shrub layer small forest shrubs, mainly bilberry *Vaccinium myrtillus* and lingonberry *Vaccinium vitis-idaea* are common [44,45]. A common feature of these spruce forests is sufficient, always-flowing moisture.

Pine forests are developed on pine terraces and interfluvial plains, formed by the common pine *Pinus silvestris* L.; lichen pine forests are widespread. Forest stands are mostly exclusively pine *Pinus silvestris*; mixtures of birch *Betula pubescens*, larch *Larix sibirica*, and spruce *Picea obovata* are less common. The herb–shrub layer is sparse, with shrubs represented by lingonberry *Vaccinium vitis-idaea*, bilberry *Vaccinium myrtillus*, and crowberry *Empetrum nigrum*; lichens dominate in the ground cover, and mosses occur in shaded areas [44,45]. The dominants of the ground cover are usually lichens of the genus Cladonia, such as *C. stellaris* (Opiz.) Pouzar & Vězda, *C. rangiferina* (L.) F.H.Wigg., and *C. arbuscula* (Wallr.) Flot., often noted as *Cetraria islandica* (L.) Ach. [46].

Green moss pine forests are also common in the study area. The tree layer is dominated by pine *Pinus silvestris* with a frequent mixture of spruce *Picea obovata*; less com-

mon inclusions are larch *Larix sibirica* and birch *Betula pubescens*. Undergrowth is usually in good condition, formed by pine *Betula pubescens* and spruce *Picea obovata*. There is often an admixture of other species (birch *Betula pubescens*, larch *Larix sibirica*). The ground cover is usually continuous, highly developed, and formed by common species of green branching mosses, with a small admixture of fruticose lichens and cuckoo flax *Polytrichum commune* [44,45]. Depending on the structure of the herbaceous shrub layer, two subgroups of associations are distinguished within the group of green-moss pine forests; one is a shrub-bush-green-moss pine forest [44]. In the taiga zone, lichens substantially influence the formation of forest biocenoses and are distinguished by the diversity of species [46]. To date, 429 lichen species have been identified in the southern and Middle Taiga of the Komi Republic [46]. Under conditions of a cold humid climate, on the parent rocks of glacial genesis prevailing in the taiga of the European part of Russia, zonal soils are formed in autonomous positions—podzols, podzolic, etc. [47].

### 2.2. Material Collection and Processing Methods

Observations were conducted in pine communities (lichen–green moss pine forest, white moss–lingonberry pine forest, lingonberry–heather pine forest, bilberry pine forest) and spruce communities (bilberry spruce forest and green moss spruce forest). Lichen samples were also collected in willow groves in the floodplain of the Kazhym River, in thickets of bird cherry on the bank of the Sysola River, in the upper swamp, and from wooden buildings in the settlement. Populations of ground (epigeic) and epiphytic lichens of different species and life forms were examined as habitats of oribatid mites (Table 1). The species affiliation of the collected lichens was determined by Dr. L.G. Biazrov. A total of 35 lichen species were examined. Lichen taxonomy is given according to [48]; life forms are given according to the Golubkova and Biazrov classification [49].

**Table 1.** Species composition and life forms of surveyed lichens.

| Life Forms of Lichens | Lichen Species |
| --- | --- |
| Ground fruticose lichen | *Cetraria islandica* (L.) Ach. (1, 2, 3, 4, 5, 6), *Cetrariella delisei* (Bory ex Schaer.) Kärnefelt & A.Thell (1, 2, 3, 4), *Cladonia arbuscula* (Wallr.) Flot. (1, 2, 3, 4, 5), *Cladonia mitis* Sandst. (1, 2, 3), *Cladonia rangiferina* (L.) F.H.Wigg. (1, 2, 3, 4, 5, 6), *Cladonia stellaris* (Opiz) Pouzar & Vězda (1, 2, 3, 4, 5, 6), *Cladonia crispata* (Ach.) Flot. var. *crispata* (1, 2, 3, 5, 6), *Cladonia digitata* (L.) Hoffm. (1, 5, 6), *Cladonia fimbriata* (L.) Fr. (1, 2, 5), *Cladonia gracilis* (L.) Willd. subsp. *gracilis* (1, 2, 3), *Cladonia uncialis* (L.) Weber ex F.H. Wigg. (1, 2, 3), *Stereocaulon* sp. (1, 2, 3) |
| Ground foliose lichen | *Peltigera aphthosa* (L.) Willd. (1, 4, 5), *Peltigera canina* (L.) Willd. (5, 6, 7), *Peltigera leucophlebia* (Nyl.) Gyeln. (1, 5, 6) |
| Epiphytic fruticose lichen | *Bryoria fuscescens* (Gyeln.) Brodo & D.Hawksw. (1, 2, 3, 4, 5, 6), *Evernia mesomorpha* Nyl. (3, 4, 5), *Ramalina calicaris* (L.) Fr. (3, 4, 5, 6), *Usnea subfloridana* Stirt. (3, 4, 5, 6) |
| Epiphytic foliose lichen | *Parmeliopsis ambigua* (Wulfen) Nyl. (3, 4, 5, 6, 8, 9), *Hypogymnia physodes* (L.) Nyl. (1, 2, 3, 4, 5, 6, 11), *Leptogium saturninum* (Dicks.) Nyl. (6, 9), *Lobaria pulmonaria* (L.) Hoffm. (4, 5, 9), *Melanohalea olivacea* (L.) O.Blanco et al. (8, 9), *Melanohalea septentrionalis* (Lynge) O.Blanco et al. (5, 8, 9), *Parmelia sulcata* Taylor (5, 8, 9, 10), *Platismatia glauca* (L.) W.L. Culb. & C.F. Culb. (5, 6), *Tuckermannopsis chlorophilla* (Willd.) Hale (5, 6, 8, 9), *Vulpicida pinastri* (Scop.) J.E. Mattsson & M.J. Lai (1, 3, 5) |
| Epiphytic crustose lichen | *Chaenotheca chrysocephala* (Turner ex Ach.) Th. Fr. (5, 6), *Lepraria incana* (L.) Ach. (5, 6), *Mycoblastus sanguinarius* (L.) Norman (3, 4), *Lepra albescens* (Huds.) Hafellner (5, 6), *Phaeophyscia ciliata* (Hoffm.) Moberg (8), *Physconia distorta* (With.) J.R. Laundon (4, 5) |

Note: The taxonomy of lichens is given according to: Westberg, M., Moberg, R., Myrdal, M., Nordin, A. & Ekman, S. 2021. Santesson's Checklist of Fennoscandian Lichen-Forming and Lichenicolous Fungi. Uppsala University: Museum of Evolution [48]. Community: 1—lichen–green moss pine forest, 2—white moss–lingonberry pine forest, 3—lingonberry–heather pine forest, 4—bilberry pine forest, 5—bilberry spruce forest, 6—green moss spruce forest, 7—small-grass meadow, 8—willow groves in the floodplain of the Kazhym River, 9—thickets of bird cherry on the bank of the Sysola River, 10—wooden buildings in the settlement, 11—upper swamp.

The material was collected in July–August 1989–1992 and then in July–August 1998. Epiphytic samples were collected at a height of 1.5–2.0 m from pine trunks in pine forests and from trunks and branches of spruces in spruce forests; epiphyte samples from 10 trees constituted an average sample. Samples of ground lichens, 100 cm$^2$ each, were collected in 10 replicates from each plant community in 1989, 1992, and 1998. A total of 180 ground lichen samples and 48 epiphytic lichen samples, of 2 L each, were analyzed. Oribatids were extracted using Tullgren funnels [50]. More than 60,000 specimens of oribatid mites were identified. To determine the species affiliation of the animals, permanent micropreparations were made using the Faure-Berlese liquid [50]. Identification of oribatid species was

performed according to [51]. The classification of life forms of oribatid mites is given according to Krivolutsky [50]. The collections included varying species: inhabitants of the soil surface and upper horizons of the litter (epigeic), inhabitants of the litter layer (hemiedaphic), inhabitants of small soil holes (euedaphic), eurybionts, and hydrobionts, as well as non-specialized species.

The taxonomy of oribatid mites is given according to the classification of L.S. Subias [52]. Synonyms in the species list are given when the species has been listed in earlier publications under a different name. Types of longitudinal (global) distribution of species are given according to the data of L.S. Subias [52]. Based on the type of longitude distribution, cosmopolitan, semi-cosmopolitan, Holarctic, Palaearctic, and European species were distinguished. The geographical distribution of species was analyzed using [53–72].

## 3. Results and Discussion

### 3.1. Taxonomic Diversity

A total of 38 families of oribatid mites were observed in lichens, 29 families in epiphytes, and 31 families in ground lichens (Appendix A). The oribatid mite families Crotoniidae, Oppiidae, Suctobelbidae, Carabodidae, and Oribatulidae had the greatest number of species in the lichens (Figure 1). The total list of lichen-dwelling oribatid mites was 87 species (Appendix A).

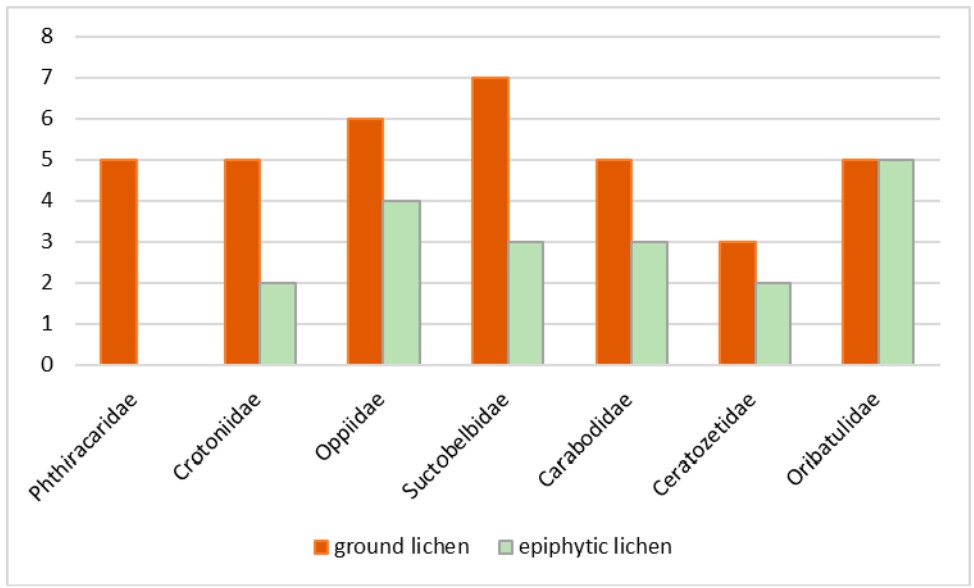

**Figure 1.** The number of species in the most common families of oribatid mites and inhabitants of ground and epiphytic lichens.

Five species of the family Crotoniidae were found in lichens, of which three species were present only in ground lichens and were not found in epiphytes; these species were *Camisia* (*C.*) *borealis*, *C.* (*E.*) *lapponica*, and *Heminothrus* (*H.*) *longisetosus.* The species, which were the most common in lichens, were *C.* (*C.*) *biurus* (in seven species of lichens) and *H.* (*H.*) *longisetosus* (in four species of lichens). Only one species, *Camisia* (*C.*) *segnis*, was associated predominantly with epiphytes.

Eight species of the family Oppiidae were found in both ground and epiphytic lichens. The species *Graptoppia* (*Apograptoppia*) *foveolata* was characterized by the greatest diversity of habitats, as it was observed in 13 lichen species. The species *Ramusella* (*R.*) *clavipectinata*, *Rhinoppia* (*R.*) *subpectinata*, *Dissorhina ornata* s. str., *Moritzoppia* (*M.*) *unicarinata* s. str. were found only in the ground lichens; these species were not found in epiphytes. Two species were observed solely in the *Hypogymnia physodes* epiphyte, *Lauroppia maritima* s. str. and *Oppiella* (*Moritzoppiella*) *neerlandica*.

Eight species of the family Suctobelbidae were found. The species *Suctobelbella (S.) acutidens duplex* was the most common in lichens; it was observed in nine species of lichens of different life forms. Species of this family were observed predominantly in ground fruticose lichens. Two more species, *S. (S.) acutidens* s. str. and *S. (F.) subtrigona*, were found in epiphytes in addition to the already mentioned *S. (S.) acutidens duplex*.

For the family Oribatulidae, five species were found in both epiphytes and ground lichens (Figure 1). In epiphytic lichens, common species were *Phauloppia nemoralis* and *Oribatula (Z.) propinqua.* The second of these species was present only in epiphytes. Both of these species were highly abundant in epiphytes, and we assigned them to the ecological group of arboreal dominant species [41]. Seyd and Seaward [28] deemed the species *P. nemoralis*, a characteristic inhabitant of lichens. According to U.Y. Shtanchaeva [36,37], the species *O. (Z.) propinqua* was one of the dominant species in the lichen *H. physodes* in a pine forest in the Bryansk region. The species *Oribatula (Z.) frisiae*, was present in our collections in four species of epiphytes and one species of ground lichen. The species *Oribatula (O.) tibialis* s. str. and *Oribatula (Z.) exilis* s. str., were the most frequent in the ground lichens of different species. Both species were also noted in some epiphytes. The former was included in the ecological group of hemiedaphic-dominant species, with the latter added to the arboreal–hemiedaphic group [41]. The species *O. tibialis* was found in the rock and epiphytic lichens of Abkhazia [22]; *O. (Z.) exilis* has been repeatedly noted as an inhabitant of epiphytic lichens [2,31,35,37].

Of the family Carabodidae five species were identified. *Carabodes (C.) labyrinthicus* was present in 17 species of lichens, mostly epiphytic (12 species). Conversely, *C. (C.) subarcticus*, which was observed in 16 lichen species, was found mainly in ground lichens (13 species). The species *C. (C.) marginatus* was also found predominantly in ground lichens (10 species). Species of this family were highly abundant in ground lichens (*C. (C.) marginatus*, *C. (C.) subarcticus*,) and in epiphytic lichens (*C. (C.) labyrinthicus*) [41]. We classified the first two species as hemiedaphic dominant, and the third species, *C. (C.) labyrinthicus*, as arboreal dominant [41]. Species of the genus Carabodes are common inhabitants of lichens in various regions, namely Poland, Austria, Germany, Denmark, Great Britain, and Scandinavia [13,14,21,23,28,33,34]. According to Andre [15–18], *C. labyrinthicus* was a numerous species in the epiphytic lichens of southern Belgium. This species, along with *C. marginatus* and *C. subarcticus*, has been observed as a characteristic inhabitant of epiphytes in the coniferous-broadleaf forest zone [2,20,35].

The species *Trhypochthonius cladonicolus* (Trhypochthoniidae) was a characteristic inhabitant of lichens of the genera Cladonia; it was observed in seven species of ground fruticose lichens. Seyd and Seaward [28] included *T. cladonicolus* in a group of species that prefer lichens as a habitat and food source but also can be found on other plants.

The species *Phthiracarus (P.) laevigatus* and *P. (P.) longulus* (Phthiracaridae) were common in the ground lichens; each was found in seven lichen species. No species of this family were noted in epiphytes. A representative of the family Euphthiracaridae, *Euphthiracarus (E.) cribrarius*, was found in five species of ground lichens.

Two species of the family Humerobatidae were found in lichens; the first is *Diapterobates oblongus*, was found in 14 species of lichens, and 9 species were epiphytic. The second species, *Diapterobates humeralis*, was noted in eight species of lichens of different life forms that were mostly epiphytic. The first species was included in the arboricolous hemiedaphic species group; the second species was in the arboricolous-dominant group [41].

The most frequent species from the family Ceratozetidae was *Trichoribates (T.) berlesei*, which was present in 14 lichen species, of which nine were epiphytic. This species was included in the group of arboreal hemiedaphic species [41]. The species *Sphaerozetes piriformis* was found predominantly in epiphytes. The third species *Melanozetes mollicomus* was noted only in ground lichens. In Central and Eastern Europe, the species *T. berlesei* was observed in the moss–lichen cover of trees [13], epiphytic lichens [31], and tree bark [14]. According to

Andre, in Belgium, *T.* (*T.*) *berlesei* (called *Trichoribates trimaculatus*) was one of the numerous species in foliose and fruticose epiphytes [15–18].

The species *Ceratoppia quadridentata* (Ceratoppiidae) was present in 20 species of lichens of different life forms, both ground and epiphytic. This species was most frequently found in foliose epiphytes and in ground fruticose species.

Some eurytopic species were frequently found in lichens. The cosmopolitan species *Tectocepheus velatus* s. str. (Tectocepheidae) was observed in 16 lichen species, both ground and epiphytic. *Scheloribates* (*S.*) *laevigatus* s. str. (Scheloribatidae) was noted in 19 species of lichens, 14 species were ground lichens, and five species were epiphytes.

Two subspecies of *Adoristes* (*A.*) *ovatus*—*A.* (*A.*) *ovatus* s. str. and *A.* (*A.*) *ovatus poppei* (Liacaridae) were found in lichens. The first subspecies was observed only in ground fruticose lichens. The second subspecies was much more frequent in lichens; it was noted in 10 species of lichens, with eight species of ground fruticose lichens. Thus, species of the genus *Adoristes* were associated with fruticose lichens. *A. ovatus poppei* was included in the group of hemiedaphic dominant species [41]. According to Niedbala [14], *A. poppei* was found mainly in forest litter, and sometimes on tree trunks.

Two subspecies of *Eueremaeus oblongus* (Eremaeidae) were found in both ground and epiphytic lichens. The subspecies *E. oblongus silvestris* was noted in seven lichen species, four species were ground fruticose, and three species epiphytes. The second subspecies *E. oblongus* s. str. was observed in four species of lichens, two of which were foliose epiphytes, and two species of fruticose ground lichens. We included *E. oblongus silvestris* in the few hemiedaphic species groups [41]. According to Andre's observations, *E. oblongus* is highly abundant in foliose and fruticose epiphytes in Belgium [16,17]; it is often found in epiphytes in Germany [13] and is a typical wood-dwelling species in Abkhazia [22]. The species *Pergalumna* (*P.*) *nervosa* s. str. (Galumnidae) was also frequently found in lichens. It was represented in 11 species of lichens, comprising mostly ground fruticose lichens and foliose epiphytes.

Some families and species were noted only in epiphytes. Specific for epiphytes were families Micreremidae and Licneremaeidae and species from these families *Micreremus brevipes* and *Licneremaeus licnophorus*, respectively. Only in epiphytes species *Furcoppia* (*M.*) *dentata* (Astegistidae), *Cymbaeremaeus cymba* (Cymbaeremaeidae), *Jacotella frondeus* (Gymnodamaeidae), *Ameronothrus oblongus* (Ameronothridae), *Mycobates* (*Calyptozetes*) *tridactylus* (Punctoribatidae), *Liebstadia* (*L.*) *humerata* (Liebstadiidae) were found. All these species were rare in collections, with occasionally singular finds. We classified some of these species (*F. dentata*, *M. tridactylus*, *C. cymba*, and *M. brevipes*) as arboreal species [41].

### 3.2. Life Forms of Oribatid Mites

Both in ground and epiphytic lichens of different life forms, the epigeic species of oribatid mites are predominated by the number of species—inhabitants of the soil surface and upper horizons of the litter, according to the classification of D.A. Krivolutsky [50] (Figure 2). Their proportion was the smallest in foliose epiphytes. Eurybionts were second in abundance of species, and their share increased significantly in crustose epiphytes. In crustose epiphytes, the share of species inhabitants of small soil holes (euedaphic species) decreased correspondingly. The inhabitants of the litter layer (hemiedaphic) were completely absent in the fruticose epiphytes. The largest number of species of this life form was noted in fruticose ground lichens. The collections included one hydrobiontic species (*Ameronothrus oblongus*), which was noted in the *Hypogymnia physodes* (foliose epiphyte). There is also one non-specialized species (*Liochthonius* (*L.*) *lapponicus*) which was present in *Parmelia sulcata* (foliose epiphyte), and *Cetrariella delisei* (ground fruticose lichen).

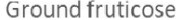

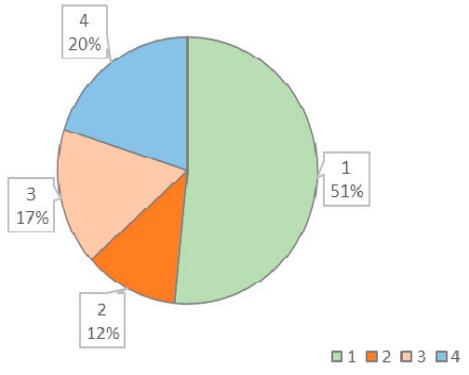

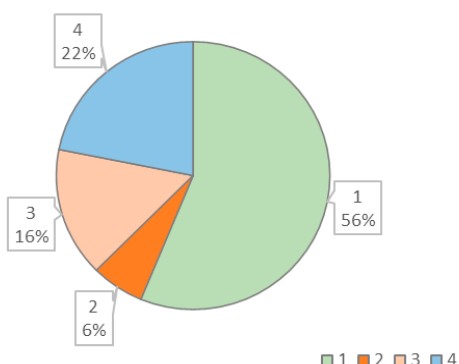

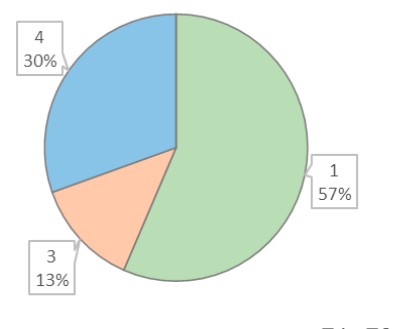

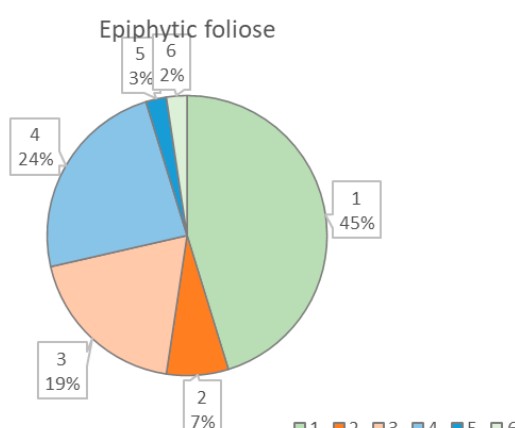

**Figure 2.** *Cont.*

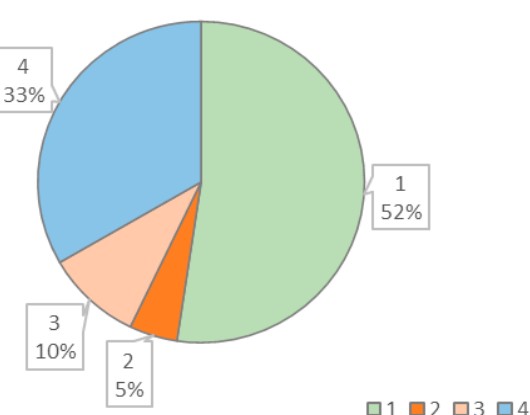

**Figure 2.** Abundance (%) of life forms of oribatid mites in lichens (1—epigeic, 2—hemiedaphic, 3—euedaphic, 4—eurybionts, 5—hydrobionts, 6—non-specialized).

### 3.3. Zoogeographic Structure of the Fauna

In the fauna of oribatid mites associated with lichens, the species with a Holarctic type of distribution were most common (Figure 3), which is characteristic of the fauna of the taiga–tundra zone of northern European Russia [69,70]. In epiphytic lichens, the proportion of Holarctic species was higher than in ground lichens. In contrast, the share of palaearctic species was smaller in epiphytes compared to ground lichens (22% and 25%, respectively). The proportion of widespread species, i.e., cosmopolites and semi-cosmopolites (*Oppiella* (*O.*) *nova s.* str., *T. velatus s.* str., *T. velatus sarekensis*, *Atropacarus* (*A.*) *striculus s.* str., *Trhypochthonius tectorum s.* str., *C.* (*C.*) *segnis*, *R.* (*R.*) *clavipectinata*, *Quadroppia* (*Q.*) *quadricarinata*, *Hemileius* (*H.*) *initialis*, *S.* (*S.*) *laevigatus* s. str., *S.* (*S.*) *pallidulus latipes*), was 12.64%, which is characteristic of northern faunas [69,70]. For comparison, in the tundra communities of the Yugorsky Peninsula, cosmopolites and semi-cosmopolites accounted for 12.5% of the total species list [71].

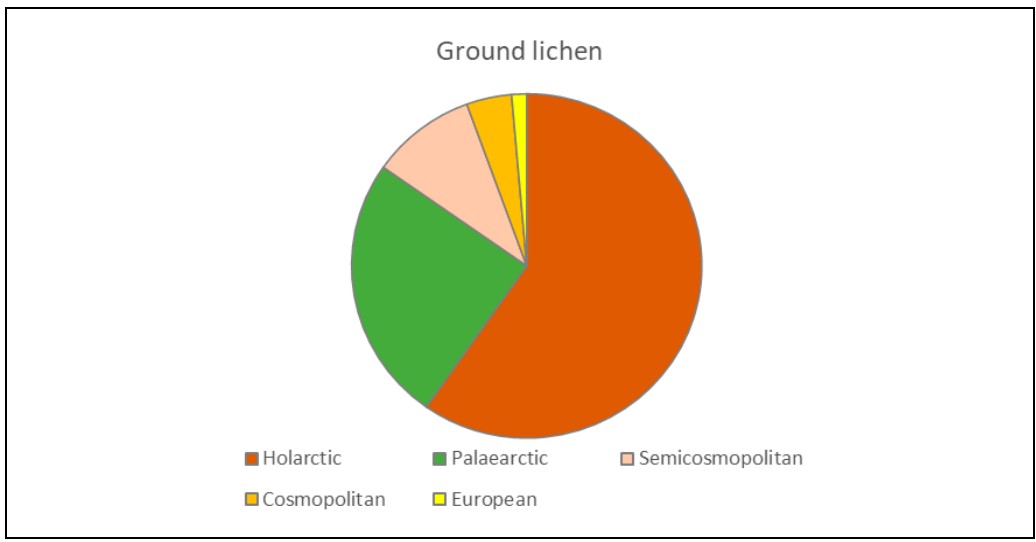

**Figure 3.** *Cont.*

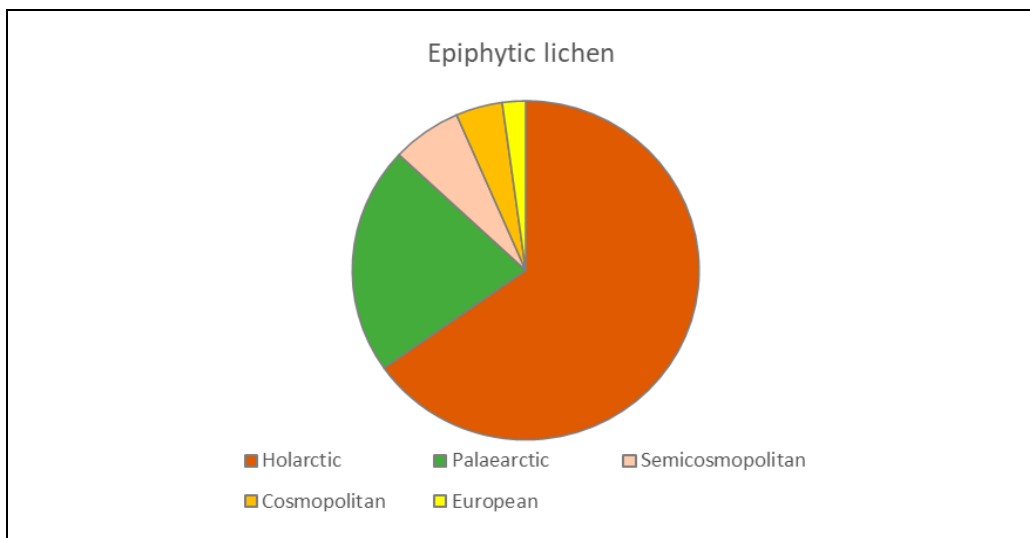

**Figure 3.** Zoogeographic structure of oribatid mite fauna.

Several groups of species can be distinguished by the nature of their distribution.

### 3.3.1. Boreal–Alpine Species

A complex of species is distinguished in the composition of the fauna, for which L.S. Subias [52] indicates a boreal–alpine distribution, these are the species *Camisia* (*C.*) *borealis* (Thorell, 1871), *Camisia* (*Ensicamisia*) *lapponica* (Trägårdh, 1910), *Melanozetes mollicomus* (Koch, 1839). These species were noted in the mainland part of the tundra zone [70]. They were common in the Svalbard Archipelago [65–67].

The Holarctic polyzonal species *Camisia* (*C.*) *borealis* was found on Vaygach Island [72], in the Polar Urals [64], on Novaya Zemlya, and Spitsbergen [65,66]. We observed the Holarctic temperate species *M. mollicomus* in tundra communities of the Pai Khoi Range [71]. It is also widespread in northern Scandinavia, the Kola Peninsula, the Northern Urals, and the Taymyr [50,53,54,69]. In northeast European Russia, the species has been noted in taiga forests [68,73]. The species *M. mollicomus* has a circumpolar distribution [74].

### 3.3.2. Species Rare to the Taiga Zone

Of interest is a group of species found in lichens that are not widely widespread in the north; these can be considered rare in the taiga zone. These species are *Furcoppia* (*Mexicoppia*) *dentata*, *Cymbaeremaeus cymba*, *Scapheremaeus palustris*, *Micreremus brevipes*, *Licneremaeus licnophorus*, *Ameronothrus oblongus*, *Liebstadia* (*L.*) *humerata*, *Diapterobates dubinini*, *Diapterobates oblongus*.

The species *F.* (*M.*) *dentata*, *C. cymba*, *S. palustris*, *M. brevipes*, *L. licnophorus*, *A. oblongus*, and *L.* (*L.*) *humerata* in the taiga zone of northeast European Russia were observed only in epiphytic lichens; they were not found in ground lichens and soil. We included these species in the ecological group of arboreal few species [41]. We also included the species *Jacotella frondeus* in this group. This species has not been previously noted in Russia [50]. Subias defines its distribution as European [52].

Species *Oribatula* (*Z.*) *frisiae* (Oudemans, 1900), *Oribatula* (*Z.*) *propinqua* (Oudemans, 1902), and *Phauloppia nemoralis* (Berlese, 1916), which are distributed in lower latitudes, are also rare in the North. The species *P. nemoralis*, which we previously noted as abundant in epiphytes [41], is not characteristic of northern latitudes. For Northern European Russia, the species is first mentioned in our studies as an inhabitant of epiphytic lichens [39,40]. It is widespread in areas of broad-leaved and coniferous broad-leaved forests [53]. In the Bryansk Region, U. Shtanchaeva [36,37] also found this species in pine forests in epiphytic lichens. In Norway, several specimens of this species were found on oak trees [75]. The same type of distribution is characteristic of species *Oribatula* (*Z.*) *propinqua* and *Oribatula* (*Z.*) *frisiae*.

Species with a more "southern" distribution also are *Diapterobates dubinini, Diapterobates oblongus*, and *Cymbaeremaeus cymba*. The species *C. cymba* habited the crowns of oaks in Norway along with *Carabodes labyrinthicus, Oribatula (Z.) exilis*, and some other species of oribatids [75].

Palaearctic species *Eueremaeus oblongus silvestris* in the European part of Russia is mainly found in the zone of broad-leaved and coniferous-broad-leaved forests [50,68]; it has rarely been found in the tundra zone. Specifically, it was found in the Polar Urals [64]; in Siberia, this species was observed in the taiga forests and Altai [54].

Thus, in the species composition of oribatid mites that inhabited lichens in the taiga forests, along with widely distributed species and species of the "northern complex", there were species that are common in more southern latitudes, i.e., species of the so-called "southern complex".

## 4. Conclusions

In ground lichens and epiphytic lichens of 35 species, 87 species of oribatid mites from 38 families were noted: in ground lichens, 72 species from 31 families were observed, and in epiphytes, 46 species from 29 families were found. The leading families according to the number of species in ground lichens were Crotoniidae, Carabodidae, Oppiidae, and Suctobelbidae; in epiphytic lichens, the family Oribatulidae was most common. The Micreremidae and Licneremaeidae families were specific for epiphytic lichens.

A complex of species characteristic of epiphytic lichens as habitats of oribatid mites have been identified, these are *Carabodes labyrinthicus, Oribatula (Z.) propinqua, Phauloppia nemoralis, Micreremus brevipes, Licneremaeus licnophorus, Furcoppia (Mexicoppia) dentata, Cymbaeremaeus cymba*. Only in epiphytes, rarely, the species were also *Jacotella frondeus, Ameronothrus oblongus, Mycobates (Calyptozetes) tridactylus* and *Liebstadia humerata*.

Characteristic of ground lichens are thhhe species *Trhypochthonius cladonicolus, Carabodes marginatus, Carabodes subarcticus*. We identified species that were often found in both ground and epiphytic lichens; these are species *Eueremaeus oblongus* s. str., *E. oblongus silvestris, Ceratoppia quadridentata, Adoristes ovatus poppei, Graptoppia (Apograptoppia) foveolata, Suctobelbella acidens duplex, Tectocepheus velatus, Trichoribates berlesei, Chamobates pusillus, Diapterobates oblongus, Oribatula tibialis, Oribatula (Z.) exilis, Scheloribates laevigatus, Neoribates aurantiacus, Pergalumna nervosa*.

The zoogeographic structure of the fauna is dominated by widespread Holarctic species, which is characteristic of the oribatid mite fauna of the taiga-tundra zone. Along with the zonal features of the fauna, specific features of the oribatid mite fauna associated with lichens were observed. Some oribatid mite species occurring in epiphytic lichens are rare in the North, such as *Oribatula (Z.) frisiae, O. (Z.) propinqua, Phauloppia nemoralis, L. licnophorus, F. (M.) dentata, C. cymba, S. palustris, A. oblongus*, and *L. humerata*, and one species was observed in the taiga zone for the first time (*J. frondeus*). Thus, epiphytic lichens house a complex of oribatid mite species that can be called "conditionally southern".

**Funding:** The work was carried out on the theme of the State Assignment of the Institute of Biology of the Federal Research Center of the Komi Scientific Center of the Ural Branch of the Russian Academy of Sciences "Fauna diversity and spatial and ecological structure of the animal population of the European North-East of Russia and adjacent territories under conditions of environmental change and economic development", registration No. 122040600025-2.

**Institutional Review Board Statement:** Not applicable.

**Data Availability Statement:** Not applicable.

**Acknowledgments:** The author thanks L.G. Biazrov for the identification of lichens and T.N. Pystina for consulting on the taxonomy of lichens. The author is grateful to L.S. Subias and U.Ya. Shtanchaeva for the sources provided and to V.A. Kanev (ground lichens) and T.G. Agaev (epiphytic lichens) for the photographs provided for the graphical abstract.

**Conflicts of Interest:** The author declares no conflict of interest.

## Appendix A

**Table A1.** Checklist of oribatid mites, inhabitants of ground and epiphytic lichens in the taiga zone of the European North-East of Russia *.

---

Brachychthonioidea Thor, 1934

---

Brachychthoniidae Thor, 1934

---

1. *Liochthonius* (*Liochthonius*) *lapponicus* (Trägårdh, 1910)
Distribution. Holarctic
*Parmelia sulcata*, *Cetrariella delisei.*

---

Euphthiracaroidea Jacot, 1930

---

Oribotritiidae Balogh, 1943

---

2. *Oribotritia* (*Oribotritia*) *berlesei* (Michael, 1898)
Distribution. Palaearctic
*Cetraria islandica, Cladonia stellaris, Cladonia digitata, Cladonia fimbriata.*

---

Euphthiracaridae Jacot, 1930

---

3. *Euphthiracarus* (*Euphthiracarus*) *cribrarius* s. str. (Berlese, 1904)
Distribution. Holarctic
*Cetraria islandica, Cladonia arbuscula, Cladonia mitis, Cladonia rangiferina, Cladonia stellaris.*

---

Phthiracaroidea Perty, 1841

---

Phthiracaridae Perty, 1841

---

4. *Atropacarus* (*Atropacarus*) *striculus* s. str. (Koch, 1835)
Distribution. Semicosmopolitan
*Cetraria islandica, Cladonia stellaris, Cladonia uncialis, Stereocaulon* sp., *Peltigera canina.*

---

5. *Hoplophthiracarus* (*Hoplophthiracarus*) *illinoisensis* (Ewing, 1909) (=*Hoploderma pavidum* Berlese, 1913)
Distribution. Semicosmopolitan
*Peltigera canina.*

---

6. *Phthiracarus* (*Phthiracarus*) *laevigatus* (Koch, 1844) (=*Hoplophora nitens* Nicolet, 1855)
Distribution. Holarctic
*Cetraria islandica, Cladonia arbuscula, Cladonia rangiferina, Cladonia stellaris, Cladonia crispata, Cladonia gracilis, Peltigera canina.*

---

7. *Phthiracarus* (*P.*) *longulus* (Koch, 1841) (=*Hoploderma boreale* Trägårdh, 1910)
Distribution. Holarctic
*Cetraria islandica, Cladonia arbuscula, Cladonia mitis, Cladonia rangiferina, Cladonia stellaris, Cladonia digitata, Peltigera aphthosa.*

---

8. *Phthiracarus* (*Archiphthiracarus*) *piger* (Scopoli, 1763)
Distribution. Holarctic
*Cetraria islandica, Cladonia stellaris.*

---

Crotonioidea Thorell, 1876

---

Trhypochthoniidae Willmann, 1931

---

9. *Trhypochthonius cladonicolus* (Willmann, 1919)
Distribution. Holarctic
*Cetraria islandica, Cladonia arbuscula, Cladonia mitis, Cladonia rangiferina, Cladonia stellaris, Cladonia digitata, Cladonia uncialis.*

---

10. *Trhypochthonius tectorum* s. str. (Berlese, 1896)
Distribution. Semicosmopolitan
*Cetraria islandica, Cladonia stellaris.*

---

Nothridae Berlese, 1896

---

11. *Nothrus silvestris* s. str. Nicolet, 1855
Distribution. Holarctic
*Hypogymnia physodes*, *Peltigera leucophlebia.*

---

Crotoniidae Thorell, 1876

---

**Table A1.** *Cont.*

---

12. *Camisia* (*Camisia*) *biurus* s. str. (Koch, 1839)
Distribution. Holarctic
*Hypogymnia physodes, Cetraria islandica, Cladonia arbuscula, Cladonia rangiferina, Cladonia stellaris, Cladonia uncialis, Peltigera aphthosa, Peltigera canina.*

---

13. *Camisia* (*C.*) *borealis* (Thorell, 1871)
Distribution. Boreoalpine. Holarctic
*Cetraria islandica, Cladonia crispata, Cladonia fimbriata.*

---

14. *Camisia* (*C.*) *segnis* (Hermann, 1804)
Distribution. Semicosmopolitan
*Leptogium saturninum, Melanohalea olivacea, Parmelia sulcata, Phaeophyscia ciliata, Physconia distorta, Cetrariella delisei, Peltigera leucophlebia.*

---

15. *Camisia* (*Ensicamisia*) *lapponica* (Trägårdh, 1910)
Distribution. Boreoalpine. Holarctic
*Cladonia stellaris, Stereocaulon* sp.

---

16. *Heminothrus* (*Heminothrus*) *longisetosus* (Willmann, 1925)
Distribution. Holarctic
*Cetraria islandica, Cladonia arbuscula, Cladonia rangiferina, Cladonia stellaris.*

---

Nanhermannioidea Sellnick, 1928

---

Nanhermanniidae Sellnick, 1928

---

17. *Nanhermannia* (*Nanhermannia*) *dorsalis* (Banks, 1896) (=*Nanhermannia coronata* Berlese, 1913)
Distribution. Holarctic
*Cetraria islandica, Cladonia rangiferina, Cladonia stellaris.*

---

Gymnodamaeoidea Grandjean, 1954
Gymnodamaeidae Grandjean, 1954

---

18. *Gymnodamaeus bicostatus* (Koch, 1835)
Distribution. Holarctic
*Peltigera leucophlebia.*

---

19. *Jacotella frondeus* (Kulijev, 1979) (*Plesiodamaeus*) (=*Plesiodamaeus ornatus* Mahunka, 1979)
Distribution. Palaearctic
*Hypogymnia physodes.*

---

Damaeoidea Berlese, 1896

---

Damaeidae Berlese, 1896

---

20. *Damaeus* (*Epidamaeus*) *bituberculatus* (Kulczynski, 1902)
Distribution. Palaearctic
*Lobaria pulmonaria, Chaenotheca chrysocephala, Lepraria incana, Physconia distorta, Cetraria islandica, Cladonia arbuscula, Cladonia rangiferina, Cladonia stellaris, Peltigera leucophlebia.*
*Hypogymnia physodes.*

---

Cepheusoidea Berlese, 1896

---

Cepheusidae Berlese, 1896

---

21. *Cepheus cepheiformis* (Nicolet, 1855)
Distribution. Holarctic
*Parmelia sulcata, Peltigera aphthosa.*

---

Gustavioidea Oudemans, 1900

---

Astegistidae Balogh, 1961

---

22. *Furcoppia* (*Mexicoppia*) *dentata* (Willmann, 1950) (*Cultroribula*)
Distribution. Holarctic
*Bryoria fuscescens, Usnea subfloridana, Hypogymnia physodes.*

---

**Table A1.** *Cont.*

| |
|---|
| 23. *Furcoribula furcillata* (Nordenskiöld, 1901)<br>Distribution. Holarctic<br>*Cladonia rangiferina.* |
| Ceratoppiidae Grandjean, 1954 |
| 24. *Ceratoppia quadridentata* (Haller, 1882)<br>Distribution. Holarctic<br>*Bryoria fuscescens, Evernia mesomorpha, Ramalina calicaris, Hypogymnia physodes, Leptogium saturninum, Lobaria pulmonaria, Melanohalea septentrionalis, Parmelia sulcata, Platismatia glauca, Tuckermannopsis chlorophilla, Chaenotheca chrysocephala, Lepraria incana, Lepra albescens, Physconia distorta, Cetraria islandica, Cetrariella delisei, Cladonia rangiferina, Cladonia stellaris, Peltigera canina, Peltigera leucophlebia.* |
| Liacaridae Sellnick, 1928 |
| 25. *Adoristes* (*Adoristes*) *ovatus* s. str. (Koch, 1839)<br>Distribution. Holarctic<br>*Cetraria islandica, Cladonia arbuscula, Cladonia rangiferina, Cladonia stellaris, Cladonia uncialis.* |
| 26. *Adoristes* (*A.*) *ovatus poppei* (Oudemans, 1906)<br>Distribution. Holarctic<br>*Hypogymnia physodes, Cetraria islandica, Cladonia arbuscula, Cladonia mitis, Cladonia rangiferina, Cladonia stellaris, Cladonia crispata, Cladonia uncialis, Stereocaulon* sp., *Peltigera leucophlebia.* |
| Tenuialidae Jacot, 1929 |
| 27. *Hafenrefferia gilvipes* (Koch, 1839)<br>Distribution. Palaearctic<br>*Cladonia stellaris.* |
| Eremaeoidea Oudemans, 1900 |
| Eremaeidae Oudemans, 1900 |
| 28. *Eueremaeus oblongus* s. str. (Koch, 1835)<br>Distribution. Holarctic<br>*Hypogymnia physodes, Parmelia sulcata, Cetraria islandica, Cladonia stellaris.* |
| 29. *Eueremaeus oblongus silvestris* (Forsslund, 1956)<br>Distribution. Palaearctic<br>*Leptogium saturninum, Lepraria incana, Phaeophyscia ciliata, Cetraria islandica, Cladonia arbuscula, Cladonia rangiferina, Cladonia stellaris.* |
| Oppioidea Sellnick, 1937 |
| Oppiidae Sellnick, 1937 |
| 30. *Graptoppia* (*Apograptoppia*) *foveolata* (Paoli, 1908)<br>Distribution. Holarctic<br>*Bryoria fuscescens, Usnea subfloridana Parmeliopsis ambigua, Hypogymnia physodes, Leptogium saturninum, Platismatia glauca, Phaeophyscia ciliata, Cetrariella delisei, Cladonia rangiferina, Cladonia stellaris, Cladonia uncialis, Stereocaulon* sp., *Peltigera aphthosa.* |
| 31. *Ramusella* (*Ramusella*) *clavipectinata* (Michael, 1885) (=*Oppia assimilis* Mihelčič, 1956)<br>Distribution. Semicosmopolitan<br>*Cladonia digitata.* |
| 32. *Rhinoppia* (*Rhinoppia*) *subpectinata* (Oudemans, 1900) (=*Oppia globosa* Mihelčič, 1956) (=*Oppia tuberculata* Bulanova-Zachvatkina, 1964)<br>Distribution. Holarctic<br>*Peltigera canina.* |
| 33. *Dissorhina ornata* s. str. (Oudemans, 1900)<br>Distribution. Holarctic<br>*Peltigera canina.* |
| 34. *Lauroppia maritima* s. str. (Willmann, 1929)<br>Distribution. Holarctic<br>*Hypogymnia physodes.* |
| 35. *Moritzoppia* (*M.*) *unicarinata* s. str. (Paoli, 1908)<br>Distribution. Holarctic<br>*Peltigera canina.* |

**Table A1.** *Cont.*

---

36. *Oppiella* (*Oppiella*) *nova* s. str. (Oudemans, 1902)
Distribution. Cosmopolitan
*Hypogymnia physodes, Cladonia arbuscula, Cladonia stellaris, Cladonia fimbriata.*

---

37. *Oppiella* (*Moritzoppiella*) *neerlandica* (Oudemans, 1900) (=*Dameosoma translamellatum* Willmann, 1923)
Distribution. Holarctic
*Hypogymnia physodes.*

---

Quadroppiidae Balogh, 1983

---

38. *Quadroppia* (*Quadroppia*) *quadricarinata* (Michael, 1885)
Distribution. Semicosmopolitan
*Hypogymnia physodes.*

---

Trizetoidea Ewing, 1917

---

Suctobelbidae Jacot, 1938

---

39. *Suctobelbella* (*Suctobelbella*) *acutidens* s. str. (Forsslund, 1941)
Distribution. Holarctic
*Hypogymnia physodes, Cladonia arbuscula.*

---

40. *Suctobelbella* (*S.*) *acutidens duplex* (Strenzke, 1950) (=*Suctobelba hammerae* Krivolutsky, 1965)
Distribution. Holarctic
*Bryoria fuscescens, Hypogymnia physodes, Platismatia glauca, Lepraria incana, Cetraria islandica, Cladonia arbuscula, Cladonia rangiferina, Cladonia stellaris, Cladonia uncialis.*

---

41. *Suctobelbella* (*S.*) *acutidens lobata* (Strenzke, 1950) (=*Suctobelba ornata* Krivolutsky, 1966)
Distribution. Palaearctic
*Cladonia arbuscula, Cladonia rangiferina, Cladonia stellaris, Cladonia uncialis, Stereocaulon* sp.

---

42. *Suctobelbella* (*S.*) *palustris* (Forsslund, 1950)
Distribution. Holarctic
*Cladonia arbuscula, Cladonia stellaris, Cladonia digitata, Cladonia fimbriata, Cladonia gracilis.*

---

43. *Suctobelbella* (*S.*) *singularis* (Strenzke, 1950)
Distribution. Palaearctic
*Cladonia arbuscula, Cladonia rangiferina, Cladonia stellaris, Cladonia crispata.*

---

44. *Suctobelbella* (*Flagrosuctobelba*) *forsslundi* s. str. (Strenzke, 1950)
Distribution. Palaearctic
*Cladonia rangiferina, Peltigera aphthosa.*

---

45. *Suctobelbella* (*Flagrosuctobelba*) *subtrigona* (Oudemans, 1900)
Distribution. Holarctic
*Hypogymnia physodes.*

---

46. *Suctobelbella* (*Ussuribata*) *latirostris* (Strenzke, 1950)
Distribution. Palaearctic
*Cetraria islandica, Cladonia arbuscula, Cladonia mitis.*

---

Carabodoidea Koch, 1843

---

Carabodidae Koch, 1843

---

47. *Carabodes* (*Carabodes*) *femoralis* (Nicolet, 1855)
Distribution. Palaearctic
*Cetraria islandica, Cladonia arbuscula.*

---

48. *Carabodes* (*C.*) *labyrinthicus* (Michael, 1879)
Distribution. Holarctic
*Bryoria fuscescens, Evernia mesomorpha, Parmeliopsis ambigua, Hypogymnia physodes, Leptogium saturninum, Lobaria pulmonaria, Platismatia glauca, Tuckermannopsis chlorophilla, Vulpicida pinastri, Lepraria incana, Lepra albescens, Phaeophyscia ciliata, Cetraria islandica, Cetrariella delisei, Cladonia arbuscula, Cladonia rangiferina, Cladonia stellaris.*

---

49. *Carabodes* (*C.*) *marginatus* (Michael, 1884)
Distribution. Palaearctic
*Hypogymnia physodes, Cetraria islandica, Cladonia arbuscula, Cladonia mitis, Cladonia rangiferina, Cladonia stellaris, Cladonia fimbriata, Cladonia uncialis, Stereocaulon* sp., *Peltigera canina, Peltigera leucophlebia.*

---

**Table A1.** *Cont.*

| |
|---|
| 50. *Carabodes* (*C.*) *ornatus* Štorkán, 1925 (=*Carabodes forsslundi* Sellnick, 1953)<br>Distribution. Palaearctic<br>*Peltigera leucophlebia.* |
| 51. *Carabodes* (*C.*) *subarcticus* Trägårdh, 1902<br>Distribution. Palaearctic<br>*Bryoria fuscescens, Hypogymnia physodes, Parmelia sulcata, Cetraria islandica, Cladonia arbuscula, Cladonia mitis, Cladonia rangiferina, Cladonia stellaris, Cladonia crispata, Cladonia digitata, Cladonia fimbriata, Cladonia gracilis, Stereocaulon* sp., *Peltigera aphthosa, Peltigera canina, Peltigera leucophlebia.* |
| Tectocepheoidea Grandjean, 1954 |
| Tectocepheidae Grandjean, 1954 |
| 52. *Tectocepheus velatus* s. str. (Michael, 1880)<br>Distribution. Cosmopolitan<br>*Bryoria fuscescens, Parmeliopsis ambigua, Hypogymnia physodes, Vulpicida pinastri, Lepraria incana, Cetraria islandica, Cladonia arbuscula, Cladonia rangiferina, Cladonia stellaris, Cladonia crispata, Cladonia digitata, Cladonia fimbriata, Cladonia gracilis, Cladonia uncialis, Stereocaulon* sp., *Peltigera canina, Peltigera leucophlebia.* |
| 53. *Tectocepheus velatus sarekensis* Trägårdh, 1910<br>Distribution. Cosmopolitan<br>*Cladonia rangiferina, Cladonia stellaris, Peltigera leucophlebia.* |
| Ameronothroidea Vitzthum, 1943 |
| Ameronothridae Vitzthum, 1943 |
| 54. *Ameronothrus oblongus* Sitnikova, 1975<br>Distribution. Holarctic<br>*Hypogymnia physodes.* |
| Cymbaeremaeoidea Sellnick, 1928 |
| Cymbaeremaeidae Sellnick, 1928 |
| 55. *Cymbaeremaeus cymba* (Nicolet, 1855)<br>Distribution. Palaearctic<br>*Bryoria fuscescens, Evernia mesomorpha, Hypogymnia physodes, Melanohalea olivacea, Parmelia sulcata, Platismatia glauca, Lepraria incana, Lepra albescens.* |
| 56. *Scapheremaeus palustris* (Sellnick, 1924)<br>Distribution. Holarctic<br>*Bryoria fuscescens, Hypogymnia physodes, Melanohalea olivacea, Chaenotheca chrysocephala, Cetraria islandica, Cladonia mitis, Cladonia stellaris.*<br>Hipoorden PORONOTICAE Grandjean, 1954 |
| Licneremaeoidea Grandjean, 1954 |
| Micreremidae Grandjean, 1954 |
| 57. *Micreremus brevipes* (Michael, 1888) (=*Micreremus gracilior* Willmann, 1931)<br>Distribution. Palaearctic<br>*Bryoria fuscescens.* |
| Licneremaeidae Grandjean, 1954 |
| 58. *Licneremaeus licnophorus* (Michael, 1882)<br>Distribution. Holarctic<br>*Hypogymnia physodes.* |
| Achipterioidea Thor, 1929 |
| Achipteriidae Thor, 1929 |
| 59. *Achipteria* (*Achipteria*) *coleoptrata* s. str. (Linnaeus, 1758)<br>Distribution. Holarctic<br>*Peltigera aphthosa.* |
| 60. *Achipteria* (*A.*) *acuta* Berlese, 1908 (=*Oribata nitens* Nicolet, 1855)<br>Distribution. Holarctic<br>*Cladonia rangiferina.* |

**Table A1.** *Cont.*

| |
|---|
| 61. *Parachipteria punctata* (Nicolet, 1855)<br>Distribution. Holarctic<br>*Cetraria islandica, Cladonia arbuscula, Peltigera canina.* |
| 62. *Campachipteria* (*Triachipteria*) *fanzagoi* (Jacot, 1929) (=*Parachipteria willmanni* Hammen, 1952)<br>Distribution. Holarctic<br>*Cetraria islandica, Cladonia rangiferina, Cladonia stellaris.* |
| Oribatelloidea Jacot, 1925 |
| Oribatellidae Jacot, 1925 |
| 63. *Oribatella* (*Oribatella*) *calcarata* (Koch, 1835)<br>Distribution. Holarctic<br>*Hypogymnia physodes.* |
| 64. *Oribatella* (*O.*) *reticulata* Berlese, 1916<br>Distribution. Holarctic<br>*Cladonia stellaris.* |
| Ceratozetoidea Jacot, 1925 |
| Ceratozetidae Jacot, 1925 |
| 65. *Melanozetes mollicomus* (Koch, 1839)<br>Distribution. Boreoalpine. Holarctic<br>*Cladonia arbuscula.* |
| 66. *Sphaerozetes piriformis* (Nicolet, 1855)<br>Distribution. Palaearctic<br>*Bryoria fuscescens, Ramalina calicaris, Lobaria pulmonaria, Parmelia sulcata, Chaenotheca chrysocephala, Cetraria islandica, Cetrariella delisei.* |
| 67. *Trichoribates* (*Trichoribates*) *berlesei* (Jacot, 1929) (=*Murcia trimaculata* Koch, 1835)<br>Distribution. Holarctic<br>*Bryoria fuscescens, Ramalina calicaris, Hypogymnia physodes, Lobaria pulmonaria, Melanohalea olivacea, Platismatia glauca, Tuckermannopsis chlorophilla, Chaenotheca chrysocephala, Mycoblastus sanguinarius, Lepra albescens, Cetraria islandica, Cladonia arbuscula, Cladonia rangiferina, Stereocaulon* sp. |
| Chamobatidae Thor, 1937 |
| 68. *Chamobates* (*Chamobates*) *pusillus* (Berlese, 1895) (=*Notaspis cuspidatus borealis* Trägårdh, 1902)<br>Distribution. Holarctic<br>*Hypogymnia physodes, Leptogium saturninum, Melanohalea septentrionalis, Parmelia sulcata, Lepraria incana, Cladonia arbuscula, Cladonia rangiferina, Cladonia stellaris, Cladonia gracilis, Stereocaulon* sp., *Peltigera aphthosa, Peltigera canina, Peltigera leucophlebia.* |
| Humerobatidae Grandjean, 1971 |
| 69. *Diapterobates dubinini* Shaldybina, 1971<br>Distribution. Palaearctic<br>*Cetraria islandica.* |
| 70. *Diapterobates humeralis* (Hermann, 1804)<br>Distribution. Holarctic<br>*Bryoria fuscescens, Ramalina calicaris, Hypogymnia physodes, Melanohalea septentrionalis, Vulpicida pinastri, Physconia distorta, Cetraria islandica, Cladonia stellaris.* |
| 71. *Diapterobates oblongus* (L. Koch, 1879)<br>Distribution. Palaearctic<br>*Bryoria fuscescens, Usnea subfloridana, Hypogymnia physodes, Leptogium saturninum, Lobaria pulmonaria, Melanohalea septentrionalis, Parmelia sulcata, Platismatia glauca, Lepraria incana, Cetraria islandica, Cladonia rangiferina, Cladonia stellaris, Cladonia gracilis, Peltigera canina.* |
| Punctoribatidae Thor, 1937 |
| 72. *Mycobates* (*Calyptozetes*) *tridactylus* Willmann, 1929<br>Distribution. Holarctic<br>*Bryoria fuscescens, Evernia mesomorpha* |
| Oripodoidea Jacot, 1925 |
| Oribatulidae Thor, 1929 |

**Table A1.** *Cont.*

| |
|---|
| 73. *Oribatula* (*Oribatula*) *pannonica* Willmann, 1949<br>Distribution. Palaearctic<br>*Cetraria islandica.* |
| 74. *Oribatula* (*O.*) *tibialis* s. str. (Nicolet, 1855)<br>Distribution. Holarctic<br>*Hypogymnia physodes, Lobaria pulmonaria, Melanohalea septentrionalis, Parmelia sulcata, Chaenotheca chrysocephala, Mycoblastus sanguinarius, Cetraria islandica, Cladonia arbuscula, Cladonia rangiferina, Cladonia stellaris, Cladonia fimbriata, Cladonia gracilis, Peltigera aphthosa, Peltigera canina, Peltigera leucophlebia.* |
| 75. *Oribatula* (*Zygoribatula*) *exilis* s. str. (Nicolet, 1855)<br>Distribution. Holarctic<br>*Bryoria fuscescens, Evernia mesomorpha, Usnea subfloridana, Parmeliopsis ambigua, Hypogymnia physodes, Leptogium saturninum, Lobaria pulmonaria, Parmelia sulcata, Vulpicida pinastri, Chaenotheca chrysocephala, Lepraria incana, Mycoblastus sanguinarius, Physconia distorta, Cetraria islandica, Cetrariella delisei, Cladonia arbuscula, Cladonia digitata, Peltigera canina, Peltigera leucophlebia,* |
| 76. *Oribatula* (*Z.*) *frisiae* (Oudemans, 1900) (=*Zygoribatula tenuelamellata* Mihelčič, 1956)<br>Distribution. Holarctic<br>*Ramalina calicaris, Usnea subfloridana, Platismatia glauca, Mycoblastus sanguinarius, Cetraria islandica.* |
| 77. *Oribatula* (*Z.*) *propinqua* (Oudemans, 1902)<br>Distribution. Palaearctic<br>*Bryoria fuscescens, Evernia mesomorpha, Usnea subfloridana, Hypogymnia physodes, Lobaria pulmonaria, Melanohalea olivacea, Chaenotheca chrysocephala, Lepra albescens.* |
| 78. *Phauloppia nemoralis* (Berlese, 1916)<br>Distribution. European<br>*Bryoria fuscescens, Evernia mesomorpha, Usnea subfloridana, Hypogymnia physodes, Lobaria pulmonaria, Melanohalea septentrionalis, Parmelia sulcata, Platismatia glauca, Vulpicida pinastri, Lepraria incana, Mycoblastus sanguinarius, Phaeophyscia ciliata, Physconia distorta, Cetraria islandica, Cladonia rangiferina, Cladonia stellaris, Cladonia fimbriata, Cladonia uncialis.* |
| Hemileiidae Balogh *et* P. Balogh, 1984 |
| 79. *Hemileius* (*Hemileius*) *initialis* (Berlese, 1908) (=*Scheloribates confundatus* Sellnick, 1928)<br>Distribution. Semicosmopolitan<br>*Cladonia rangiferina, Cladonia fimbriata, Stereocaulon* sp. |
| Liebstadiidae Balogh *et* P. Balogh, 1984 |
| 80. *Liebstadia* (*Liebstadia*) *humerata* Sellnick, 1928<br>Distribution. Holarctic<br>*Hypogymnia physodes.* |
| 81. *Liebstadia* (*L.*) *pannonica* s. str. (Willmann, 1951) (=*Protoribates novus* Willmann, 1953)<br>Distribution. Holarctic<br>*Hypogymnia physodes, Parmelia sulcata.* |
| Scheloribatidae Grandjean, 1933 |
| 82. *Scheloribates* (*Scheloribates*) *laevigatus* s. str. (Koch, 1835)<br>Distribution. Semicosmopolitan<br>*Bryoria fuscescens, Parmeliopsis ambigua, Hypogymnia physodes, Parmelia sulcata, Lepraria incana, Cetraria islandica, Cladonia arbuscula, Cladonia mitis, Cladonia rangiferina, Cladonia stellaris, Cladonia crispata, Cladonia digitata, Cladonia fimbriata, Cladonia gracilis, Cladonia uncialis, Stereocaulon* sp., *Peltigera aphthosa, Peltigera canina, Peltigera leucophlebia.* |
| 83. *Scheloribates* (*S.*) *pallidulus latipes* (Koch, 1844)<br>Distribution. Semicosmopolitan<br>*Cetraria islandica.* |
| Parakalummidae Grandjean, 1936 |
| 84. *Neoribates* (*Neoribates*) *aurantiacus* (Oudemans, 1914)<br>Distribution. Holarctic<br>*Bryoria fuscescens, Hypogymnia physodes, Lobaria pulmonaria, Melanohalea septentrionalis, Tuckermannopsis chlorophilla, Cetraria islandica, Cetrariella delisei, Cladonia arbuscula, Cladonia rangiferina, Cladonia stellaris, Cladonia uncialis, Peltigera leucophlebia.* |

**Table A1.** *Cont.*

| |
|---|
| 85. *Neoribates* (*N.*) *roubali* (Berlese, 1910)<br>Distribution. Palaearctic<br>*Cladonia stellaris.* |
| Galumnoidea Jacot, 1925 |
| Galumnidae Jacot, 1925 |
| 86. *Galumna* (*Galumna*) *lanceata* (Oudemans, 1900) (=*Zetes dorsalis* Koch, 1835)<br>Distribution. Palaearctic<br>*Cetraria islandica*, *Cladonia rangiferina* |
| 87. *Pergalumna* (*Pergalumna*) *nervosa* s. str. (Berlese, 1914)<br>Distribution. Holarctic<br>*Hypogymnia physodes*, *Parmelia sulcata*, *Vulpicida pinastri*, *Cetraria islandica*, *Cetrariella delisei*, *Cladonia arbuscula*, *Cladonia mitis*, *Cladonia stellaris*, *Cladonia crispata*, *Peltigera aphthosa*, *Peltigera canina.* |

* Note. The global distribution of oribatid mites follows Subias, 2022 [52].

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
