# Peer review of "Lichen-Associated Oribatid Mites in the Taiga Zone of Northeast European Russia: Taxonomical Composition and Geographical Distribution of Species"

_diversity, doi:10.3390/d15050599_

Round 1
Reviewer 1 Report
The paper is a straightforward and sound traditional description of mite distribution in the Komi area and their occurrence in different lichens of the Taiga.It is a paper written by an expert, results of a long work in oribatid mites of the Arctic and Subarctic. It is a neat and coherent paper and surely merits the publication.
This specific study is a follow up to an earlier paper from 2020 (Oribatid Mites as Inhabitants of Lichens in the Taiga Zone of Northeastern Europe: Biotopic Association and Ecological Groups of Species). Additional research from the same area and collected by the same methods is added. The ecological classifications have been established in the earlier paper, thus the present paper is not to develop a new method, but it gives the reults: an exhaustive and concise checklist (87 species) and sum-up of the study of oribatid mites in lichens.
The long description of the occurrence of mites of different occurrence types (3.1. Taxonomic diversity) is long and a bit hard to read (Quality of presentation). The author might think about to shorten it by rewording and adding tables and figures.
The literature list is good and exhaustive.
In the text, one table gives the lichen species attributed to habitat types, one figure shows the number of species of the most common mite families and if they occur in either ground or epiphytic lichen and another figure shows the zoogeographic structure of the mite fauna without any reference of habitat type.
While the detailed description is a merit, the paper could do with more graphs to show the results, which remain hidden in the text and appendix. The aforementioned article could serve as example.
E.g. a table of the five ecological types and the number of species in the lichen types and additional figures similar to figure 2 with the zoogeographic structure of ground and epiphytic lichens separate

Author Response
Dear reviewer,
The author is very grateful to the respected reviewer for your attention to the manuscript, very valuable comments and corrections.
I tried to take into account all the comments and suggestions and made corrections to the manuscript. All made corrections and additions are highlighted in color.
Best regards,
author Elena N. Melekhina

Reviewer 2 Report
Dear Elena
I was exited that paper about oribatid mites appeared. New species in tundra - it is important but unfortunately quality of presentation need to be improved.
Introduction need more scientific background, paragraph two should be much longer and present most important findings from cited publications. Paragraph third (line 46-51) must be write again, for now I understood it will be a review article not a research one.
In 2.1. chapter descriptions of different types of forests are nor equal - sometime certain information appear in one and in next one no e.g. lichens names, Latin names of trees and other plants. It must be unified and expand to the same style. It will help readers and will be more scientifically sound.
lines 96-107 and table 1 it will be much easier to read and understand if information about forest formation and life forms of lichens will be combine in one table (even if different types of lichens where found in the same type of forest).
Results need to be write again. First description of species ecology is good for MSc thesis not for journal with IF. It is lack of ecological analysis. Where is data about domination, constancy, diversity index? Please calculate it (you have nice number of moss mites individuals). And compare them using statistics. You can look for inspiration to Piotrs Skubała papers.
If you want to conclude about indicator species (conclusions lines 338-348) in results part it must be proven e.g. by PCA or CA analysis.
3.2. part will be easier to read if some kind of table or graph will be included (plus figure2). Then discussion can follow.
Minor: 3.2.1. and 3.2.2. need to be unify according to journal rules.
Data are interesting and important but presentation needs lot of additional work.
Author Response
Dear reviewer,
The author is very grateful to the respected reviewer for your attention to the manuscript, very valuable comments and suggestions.
I tried to take into account all the comments and suggestions and made corrections to the manuscript. All made corrections and additions are highlighted in color.
Best regards,
author Elena N. Melekhina

Round 2
Reviewer 2 Report
Dear Elena
thank you for improvements and response.